# Putative Biomarkers for Prognosis, Epithelial-to-Mesenchymal Transition, and Drug Response in Cell Lines Representing Oral Squamous Cell Carcinoma Progression

**DOI:** 10.3390/genes16020209

**Published:** 2025-02-09

**Authors:** Mohamad Z. Hamoui, Shuaa Rizvi, Hilal Arnouk, Cai M. Roberts

**Affiliations:** 1Biomedical Sciences Program, Midwestern University, Downers Grove, IL 60515, USA; 2Department of Pathology, Midwestern University, Downers Grove, IL 60515, USA; 3Department of Pharmacology, Midwestern University, Downers Grove, IL 60515, USA

**Keywords:** oral squamous cell carcinoma, stratifin, biomarkers, epithelial to mesenchymal transition, cadherins, docetaxel

## Abstract

Background/Objectives: Oral squamous cell carcinoma (OSCC) is the most common form of head and neck cancer and accounts for over 50,000 new cancer cases annually in the United States. The survival rates are markedly different for localized OSCC versus metastatic disease, for which the five-year survival rate is only 39%. Depending on its pathology and stage at diagnosis, the treatment may involve surgery, radiation, targeted therapy, or conventional chemotherapy. However, there is an unmet need for reliable biomarkers to predict the treatment response or link therapeutic efficacy to tumor progression. We sought to assemble a panel of OSCC tumor progression biomarkers that correlated with the epithelial-to-mesenchymal transition (EMT) and the response to cytotoxic drugs. Methods: We used four cell lines that represented the stepwise progression from normal oral mucosa to dysplastic, invasive, and metastatic OSCC lesions and performed a quantitative analysis via Western blot for putative markers. EMT phenotypes were assessed using wound healing migration assays. Live cell imaging was used to assess drug effectiveness over time. Results: The expression of stratifin, a tumor suppressor gene, is inversely correlated with both tumor progression steps and the expression of the EMT marker N-cadherin. Conversely, the E-cadherin and fibronectin expression was markedly decreased in the advanced-stage OSCC lines. In addition, metastatic Detroit 562 cells exhibited resistance to cell death following docetaxel treatment and showed clear migratory behavior. Conclusions: We describe a molecular signature of advanced and drug-resistant OSCC tumors which encompasses multiple markers, warranting further investigation to establish their utility in predicting clinical outcomes and guiding the treatment options for patients afflicted with oral cancer.

## 1. Introduction

Oral squamous cell carcinoma (OSCC) is the most common form of head and neck cancer [1]. Currently, it is also the most diagnosed cancer in Southeastern Asia and Southern and Eastern Africa, while lip and oral cavity cancer is the leading cause of cancer mortality in Indian men [2]. Meanwhile, in the US, approximately 58,450 new cases were expected in 2024, with 12,230 expected deaths [3]. The use of known carcinogens such as alcohol and chewing tobacco are major risk factors for oropharyngeal cancer [4]. In addition to avoiding these, early detection is key to favorable outcomes; the overall five-year survival in the US is 69%, but this figure drops to 39% in patients with distant metastases [3].

Currently, there is no routine screening program for OSCC, but early detection can be achieved during oral exams. Importantly, leukoplakia, thin white plaques within the oral cavity, can be an early sign of dysplasia or neoplasia. Studies have shown that anywhere from 3% to 36% of leukoplakia will become malignant [5,6]. Leukoplakia is characterized by dysplastic changes of varying severity. These dysplastic cells are considered premalignant because they can potentially progress to invasive carcinoma [7]. Therefore, it is recommended that abnormal oral plaques be biopsied and histopathologically evaluated.

The prognosis for a patient with a diagnosis of OSCC depends primarily on TNM (tumor, node, and metastasis) clinical staging. Although these parameters are useful, Woolgar and Triantafyllou suggest that shortcomings exist due to the reliance on morphological features alone, the inherent limitations of staging, and misinterpretations. These pitfalls can impact patients’ prognosis [8]. Molecular biomarkers could serve as diagnostic and prognostic tools to supplement TNM staging, predict the extent of oral cancer progression for individual patients rather than populations, and help mitigate poor outcomes. Some studies of biomarkers for OSCC have already been performed. For instance, Yoshimura et al. found that Ubiquitin-Conjugating Enzyme E2S was upregulated in nine OSCC cell lines, in addition to other cancer types [9]. Conversely, Jiang et al. found that expression of tapasin was reduced in OSCC [10]. We previously showed that in line with other squamous tumors, OSCC shows the reduced expression of cornulin [11,12]. However, further work in this area is needed, and we aim to identify new combinations of biomarkers to improve OSCC diagnosis and prognosis. Our long-term goal is to combine several markers, including those examined in this study, into a panel that can supplement and enhance the predictive power of pathological examination and help differentiate advanced- from earlier-stage disease.

In order to capture the full picture of OSCC progression and biomarker expression, it is necessary to begin with precancerous lesions. Waldron and Shafer found that 42.9% of oral leukoplakias on the floor of the mouth showed some degree of epithelial dysplasia or unsuspected invasive carcinoma [13]. Clinically, it is important to be able to differentiate these steps. Furthermore, the treatment will be radically different for local disease versus distant metastasis, so it is vital to identify tumors that have spread or may soon spread to other sites. To address all of these concerns, we elected to use a panel of established cell lines representing the full spectrum of progression phenotypes, from normal to aggressive OSCC (Figure 1). Primary gingival keratinocyte (PGK) cells are the normal controls, dysplastic oral keratinocytes (DOKs) represent mild to moderate dysplasia, and SCC-25 is an established OSCC cell line representative of locally invasive carcinoma. Detroit 562 (Detroit) cells were derived from the pleural effusion of a patient with metastatic oropharyngeal squamous cell carcinoma and represent the most aggressive stage of oral cancer. While these were derived from a distinct location from that of the other OSCC lines, prior proteomic analyses have demonstrated that anatomical location makes no significant difference in terms of the overall protein expression profile of squamous head and neck tumors [14]. Therefore, we believe Detroit 562 to be an adequate model of metastatic OSCC. Additional information on the origin and characteristics of these lines is given in Appendix A.

One potential source of prognostic biomarkers is the epithelial-to-mesenchymal transition (EMT) pathway. EMT is vital to tissue remodeling and morphogenesis during normal development [15]. However, EMT is often reactivated in cancers, allowing malignant cells to migrate and invade the surrounding tissues [16]. Several signals can activate an EMT program, but among the best characterized is transforming growth factor β (TGFβ) [17,18]. Ali et al. showed that well-differentiated OSCC expressed high levels of the epithelial marker E-cadherin, while poorly differentiated tumors had reduced E-cadherin and increased N-cadherin expression. This “cadherin switch” is commonly seen in EMT and reflects the change from an anchored epithelium to motile, mesenchymal-like cells [19]. We sought to confirm this finding to validate our cell line model of OSCC progression and to expand to other markers of EMT.

In recent years, EMT has been implicated not only in metastasis but also in changes in drug response across many cancers [20]. Notably, Steinbichler and colleagues analyzed the effect of treatment using fibroblast-conditioned medium or TGFβ on OSCC cells and found an increase in mesenchymal marker expression and resistance to chemotherapy and radiation [21,22]. Ingruber et al. also found a correlation between EMT markers and platinum resistance in a model of OSCC [23,24]. How these findings will translate to the full spectrum of OSCC progression and other therapeutics will be the focus of our study.

Based on the prior literature on related tumor types, we hypothesized that two additional potential markers may be of use: stratifin (*SFN*) and glutathione S-transferase Pi (*GSTP1*). SFN was first described in 1967 as part of the larger 14-3-3 family of proteins in the mammalian brain [25]. Of the seven isoforms present in humans, only 14-3-3σ (*SFN*) possesses anti-tumor activity [26,27]. SFN has a role in suppressing cancer cell growth and metastasis and has been implicated in regulating a range of proteins involved in oncogenesis [25,26,27,28,29,30]. Its loss is implicated in esophageal and lung carcinomas and other squamous cell carcinomas [31,32,33]. Meanwhile, GSTP, known for its function as an antioxidant, has been shown to be increased in preneoplastic and dysplastic lesions of head and neck squamous cell carcinoma [34,35]. A study conducted by Yang et al. revealed an interaction between an increase in *GSTP1* and EMT in hepatocellular carcinoma (HCC). Additionally, the GSTP1-associated EMT contributes to resistance to paclitaxel, a drug closely related to docetaxel [36]. Given the links with EMT and progression in related tumor types, we hypothesized that GSTP could be another marker of advanced, drug-resistant OSCC.

In this study, we used a panel of four cell lines representing the successive steps in oral squamous cell carcinoma progression to identify putative biomarkers of the progression, EMT, and therapeutic response at each stage. To our knowledge, ours is the first study to investigate biomarker expression across the full range of OSCC progression, all the way from normal tissue to distant metastasis, using these lines.

## 2. Materials and Methods

### 2.1. Cell Lines and Reagents

Four cell lines were used in this study: PGK, DOK, SCC-25, and Detroit 562. The origins and culture conditions for all lines are given in Appendix A. The PGK, SCC-25, and Detroit 562 cells were obtained from the American Type Culture Collection (ATCC, Manassas, VA, USA). The DOKs were obtained from the European Collection of Authenticated Cell Cultures (ECACC). All of the cell lines were grown in a humidified incubator at 37 °C and 5% CO_2_ and maintained at a low passage number to limit the changeover time in the culture. All cell growth media were obtained from the ATCC and supplemented according to the recommended formulations to maintain the cell phenotypes. Equally, 1% penicillin–streptomycin (Gibco, ThermoFisher Scientific, Waltham, MA, USA) was added to the growth media to prevent bacterial contamination. Sytox dead cell stain was obtained from ThermoFisher. Docetaxel was sourced from SelleckChem (Houston, TX, USA).

### 2.2. Western Blotting

The cells were harvested and lysed in radioimmunoprecipitation assay (RIPA) buffer supplemented with protease and phosphatase inhibitors. Following centrifugation of the lysate, the protein concentration in the supernatant was determined using a bicinchoninic acid (BCA) assay (ThermoFisher). Equal masses of the protein were run on gradient polyacrylamide gels (Bio-Rad, Hercules, CA, USA) and then transferred onto polyvinylidene fluoride (PVDF) membranes. The membranes were rinsed in tris-buffered or phosphate-buffered saline with 0.5% Tween-20 (TBST or PBST, respectively) and then blocked in 5% milk in TBST/PBST. The membranes were then incubated overnight at 4 °C in the primary antibody. The following day, the blots were rinsed in TBST/PBST, incubated in the secondary antibody for 1–2 h, and rinsed again. The blots were then incubated with enhanced chemiluminescent substrate (Bio-Rad) and imaged using a ChemiDoc imager (Bio-Rad). Bio-Rad ImageLab software (Version 5.2.1) was used to quantify the blot images. The band intensities of proteins of interest were normalized to those of actin (the endogenous control, chosen due to its ubiquitous expression in all lines), and then the relative expression was calculated by setting the expression in one cell line to 1. The primary antibodies and their dilutions are listed in Appendix A. The horseradish-peroxidase-conjugated anti-mouse and anti-rabbit secondary antibodies were obtained from Invitrogen and Santa Cruz.

### 2.3. Scratch Wound Healing Assays

To investigate the migratory ability of the OSCC cell lines at different stages of progression, a wound healing assay was performed. The cells were plated into 24-well plates and allowed to incubate until they reached confluency. A P200 pipette tip was used to create a scratch down the center of each well, simulating a wound. Subsequently, the medium was removed, and the wells were gently washed twice using phosphate-buffered saline (PBS) to eliminate any floating cells. Fresh medium was then added to the wells. The 24-well plate was placed into the Celloger Mini Plus live cell imager (Curiosis, Seoul, Korea), where it was imaged every hour over a 48 h period. Time-lapse movies and still images were created to visually depict the process of wound closure.

### 2.4. Drug Response Assays

To assess the response of the OSCC cell lines to cisplatin and docetaxel, the Celloger Mini Plus live cell imager was used. The cells were plated at a density of 3000–6000 cells per well in a Nunc Edge 96-well plate (ThermoFisher) and allowed to adhere overnight. The following day, varying concentrations of the chemotherapy drugs were added to the respective wells, while PBS was added to all empty wells and to the moat areas of the plate to prevent evaporation. The 96-well plate was then placed in the Celloger Mini Plus live cell imager, and the cells were imaged every 4 h over a 72 h period. The instrument measured the percent confluence of the cells, providing a real-time assessment of the cell proliferation over time. Additionally, to quantify cell death throughout the experiment, the Sytox dead cell stain reagent was included in the wells (at 1:667 dilution). This stain fluoresced green when cells died, allowing for an analysis of the area of the green signal to determine the extent of cell death.

### 2.5. Statistics and Replication

Comparisons between multiple cell lines were analyzed using a one-way analysis of variance (ANOVA). Tukey’s test for multiple comparisons was used to compare each cell line’s mean to that of each of the others. *p* < 0.05 was considered statistically significant. All of the statistical analyses were performed using GraphPad Prism, versions 9 and 10. All experiments were repeated three times, except where otherwise noted. All graphs show the mean ± standard error (SEM).

## 3. Results

### 3.1. EMT Markers Correlate Well with OSCC Progression in the Cell Lines

We hypothesized that the SCC-25 and Detroit 562 cells, which have an invasive and metastatic phenotype, would exhibit a more mesenchymal-like gene expression profile. To test this hypothesis, we assayed the expression of three EMT markers using Western blot. Surprisingly, the expression of fibronectin (FN) was high in the PGK and DOK cells and decreased significantly in SCC-25 and Detroit 562 (Figure 2A,B). As expected, the PGKs had the highest expression of the epithelial marker E-cadherin (E-Cad), with all of the cancerous cells exhibiting significantly lower E-cad levels. There was a small upward trend in the E-cad expression from the DOKs to Detroit, however (Figure 2A,C). In contrast, the mesenchymal marker N-cadherin (N-Cad) was almost absent in the PGKs, and its expression increased steadily with OSCC progression, culminating with high expression in Detroit (Figure 2A,D).

### 3.2. All Cell Lines Close a Scratch Wound, but Detroit 562 Shows Distinct Projections

We next wanted to determine whether an increased mesenchymal character meant that our cell lines would display differences in their migratory ability. As a simple test, we conducted scratch wound assays using time-lapse imaging. We hypothesized that the DOK cells would be slow to close a scratch, while SCC-25 and Detroit would be faster to heal. Interestingly, the DOKs closed a wound in 24–30 h (Figure 3A and Appendix A). SCC-25 and Detroit had similar rates of wound closure over the course of 48 h (Figure 3A and Appendix A). Looking more closely, we observed that the Detroit cells extended long projections at the leading edge of the migrating cell front that were not observed in the other lines (Figure 3B). The DOK cells were larger than the other two lines, and the DOK and SCC-25 cells appeared to proliferate to a greater extent than Detroit, which may explain why the DOKs were often able to close the gap the fastest (Appendix A). This is supported by previous reports of doubling times of around 40 h for DOKs, 32–58 h for SCC-25, and as long as 88 h for Detroit cells [22,37,38]. Future studies will examine the ability of each line to invade through an extracellular matrix, which will better distinguish the contributions of proliferation and migration to the apparent motion of these cell populations.

### 3.3. The Expression of Stratifin and GSTP Throughout the Progression Steps of OSCC

In analyzing the expression of stratifin (SFN), there was an overall downward trend of decreasing SFN expression as the cancer progressed (Figure 4A,B). Pairwise comparisons were made between cell lines representing consecutive steps from normal to dysplasia, dysplasia to locally invasive, and locally invasive to metastatic oral cancer. There was a statistically significant difference in the SFN expression between PGKs and DOKs and between PGKs and Detroit. There was no significant difference between DOKs and SCC-25, but there was a strong downward trend between SCC-25 and Detroit. However, this change did not achieve statistical significance after correcting for the comparisons between all lines. Thus, stratifin expression appears to inversely correlate with the progression from normal to locally invasive to metastatic OSCC. In order to evaluate the utility of GSTP as a biomarker, we also performed Western blots for this protein in all four cell lines. No significant differences were observed in the GSTP expression among the cell lines, although an increasing trend was evident from PGKs to Detroit (Figure 4A,C). Taken together, our Western blot data suggest that SFN, FN, and cadherin levels together represent a promising panel of diagnostic biomarkers for OSCC staging. Additional studies in patient samples are needed to confirm this, as well as the inverse correlation between SFN expression and GSTP expression as they relate to the successive progression steps.

### 3.4. Detroit Cells Exhibit a Distinct Response to Docetaxel Versus That in the Other Lines

Given the correlation between the EMT and drug resistance in the literature, as well as the much lower survival rate of metastatic OSCC, we hypothesized that the Detroit 562 cells would be resistant to chemotherapy. To test this hypothesis, we treated the DOK, SCC-25, and Detroit cells with a ten-fold dose dilution series of docetaxel, a cytotoxic drug used clinically in the management of OSCC (range: 1 nM–10 µM). The drug response was evaluated using automated time-lapse imaging via the Celloger Mini Plus imaging system. The DOK and SCC-25 cells showed a somewhat similar reduction in cell growth and proliferation at all of the doses tested, as measured using cell confluency (Figure 5A, Appendix A). The Detroit cells also responded to all doses tested, but there was a trend toward a dose-dependent reduction in cell growth, with increased growth at 1 nM versus that at the higher doses. However, when adjusting for multiple comparisons, this trend did not achieve statistical significance, although the degree of significance between non-treated cells and higher doses did increase (Figure 5A, right panel). For each experiment, Sytox dead cell stain was added along with docetaxel. Live cells exclude the stain, while dead cells will show green fluorescence. Both the DOK and SCC-25 cells showed an extensive green signal, particularly at the 10 µM dose. However, the Detroit 562 cells showed far fewer green cells upon the 10 µM docetaxel treatment, suggesting that docetaxel could prevent proliferation but was not outright lethal in Detroit cells to the degree seen in the other lines (Figure 5B, Appendix A). Taken together, these data suggest that Detroit cells are more likely to be therapy-resistant, although in vivo studies will be necessary to evaluate the true extent of this effect. While the response to docetaxel has been evaluated in OSCC cell lines previously, we demonstrate the comparative responses over time in DOKs, SCC-25, and Detroit 562 here and their correlation with biomarker expression.

## 4. Discussion

In this study, we investigated the relationship between the biomarker expression, EMT, drug response, and disease progression in oral squamous cell carcinoma. We compared four steps of OSCC progression, from the normal oral mucosa to premalignant dysplastic, then locally invasive, and then metastatic keratinocytes, using in vitro cell models (Figure 1). Throughout this study, we utilized four cell lines as representations of the successive steps of OSCC progression. While other studies have begun to look at biomarkers in OSCC, no studies have covered the full spectrum from normal keratinocytes through to metastatic disease. The main significance of our study therefore is the full picture painted regarding the changes in the expression of biomarkers of interest throughout the course of OSCC.

We sought to tie the EMT and therapeutic response to the progression of OSCC. We expected that markers of a mesenchymal phenotype would correlate with the OSCC progression in our cell line models. Indeed, we confirmed prior reports of a “cadherin switch” from E-Cad to N-Cad expression in the four cell lines tested (Figure 2). Our other data suggest that a cadherin switch can, in combination with other markers, indicate more than the degree of tumor differentiation, as was studied previously [19]. Interestingly, the FN expression markedly decreased with OSCC progression. Fibronectin is widely considered to be a marker of mesenchymal cells, so our observation of the opposite trend in these cell lines was unexpected [39,40,41]. It may be that oral mucosa keratinocytes differ from the other tissues in which cancer EMT takes place. Additional factors such as hypoxic conditions or the presence of active epidermal growth factor/epidermal growth factor receptor (EGF/EGFR) signaling may influence FN levels and warrant further study in these lines [40,42]. Fibronectin also has a role in tissue remodeling, which may explain its expression in normal cells [43]. In any event, it will be of great interest to see whether this trend is recapitulated in OSCC patient samples, which we plan to examine in the next phase of our studies.

In the wound healing assays, we observed projections in the Detroit 562 cells that were absent in the other two cell lines, suggesting these cells more readily migrated. However, all three lines tested were able to close a scratch wound (Figure 3). Future studies will use transwell assays with and without an artificial matrix in order to distinguish the relative contributions of proliferation and migration to the results we have shown here and to evaluate each line’s ability to invade through the extracellular matrix.

Stratifin is expressed in epithelial keratinocytes and possesses tumor suppressing activity [26,27]. Its expression in keratinocytes appears to be epigenetically regulated, and the SFN gene is hyper-methylated in tumors, leading to the downregulation of SFN and further tumorigenesis [33,44]. The increased expression of SFN may also predict a poor prognosis in several cancers, including nasopharyngeal, esophageal, oral, and breast carcinoma [31,32,33,44,45,46]. The findings from our study showed that there was an overall decrease in the SFN expression as OSCC progressed from the normal oral mucosa to premalignant lesions and then to locally invasive and metastatic phenotypes (Figure 4). Clinically, the change from a normal to premalignant cell type may serve as a diagnostic marker of the potential of developing cancer [47]. Regarded as a tumor suppressor, the SFN expression has been shown to be downregulated in several carcinomas, including lung, breast, and esophageal cancer [32,33,48]. Notably, decreased SFN expression was also found to be correlated with lymph node metastasis. This emphasizes the utility of SFN as a prognostic factor since a poor prognosis in esophageal squamous cell carcinoma patients is associated with lymph node metastases [49]. Consistent with these findings, we show SFN to be downregulated when comparing the normal oral mucosa to oral cancer, suggesting that it might serve as both a diagnostic and prognostic factor for oral carcinoma patients. Some have argued that the role of SFN may be a “double-edged sword”, in that its increased expression causes resistance to radiation and anticancer agents, but this does not appear to be the case in our model [45].

Furthermore, while the trend seen in GSTP expression in our lines did not achieve statistical significance, it would be of interest to analyze the GSTP expression in a large cohort of patient tumor samples. It may also be that GSTP has a place in a panel of markers. By itself, it is not sufficient to discriminate between OSCC stages, but in a panel of markers, it may have predictive power.

Given the poor prognosis of advanced-stage disease, we hypothesized that the EMT and progression would both correlate with reduced therapeutic responses. Therefore, we analyzed the survival of DOK, SCC-25, and Detroit cells following their treatment with docetaxel (Figure 5). While the cell confluence was similar across all cell lines, especially at high doses, the Detroit cells appeared to be more resistant to cell death, as indicated by reduced Sytox staining (Figure 5B). Therefore, we conclude that progression does lead to a reduced response to docetaxel, which tracks with the reduced survival rates in patients with advanced OSCC. Tumors that have spread far beyond the primary site and that can remain dormant during therapy will naturally be more difficult to eliminate.

The key advantages of the cell line models we chose were their homogeneity and their representation of the full spectrum of oropharyngeal tumor development. However, we acknowledge a key limitation of this study: cell lines do not reflect the complexity of the tumor microenvironment, as samples from tissue biopsies would. This study represents a pilot or discovery phase. To validate the findings of this study, an immunohistochemical analysis of the proteins of interest in a large cohort of OSCC patients is planned to establish the diagnostic and prognostic utility of these potential biomarkers in a clinical setting.

## 5. Conclusions

As summarized in the schematic depicted in Figure 6, we show that the SFN expression decreases from normal to precancerous lesions and from early- to advanced-stage OSCC. SFN expression is correlated with E-Cad and FN expression and inversely proportional to N-Cad expression in four cell lines representing the course of OSCC progression. The Detroit 562 cells exhibited less cell death versus that in the other lines in response to docetaxel treatment. Therefore, SFN, FN, and E-Cad/N-Cad expression represents a potential panel of biomarkers for accurately staging OSCC tumors and providing information about progression and the likely therapeutic responses throughout treatment. Validation of this biomarker panel in patient samples will be required before these findings can be brought to bear on future cases.

## Figures and Tables

**Figure 1 genes-16-00209-f001:**
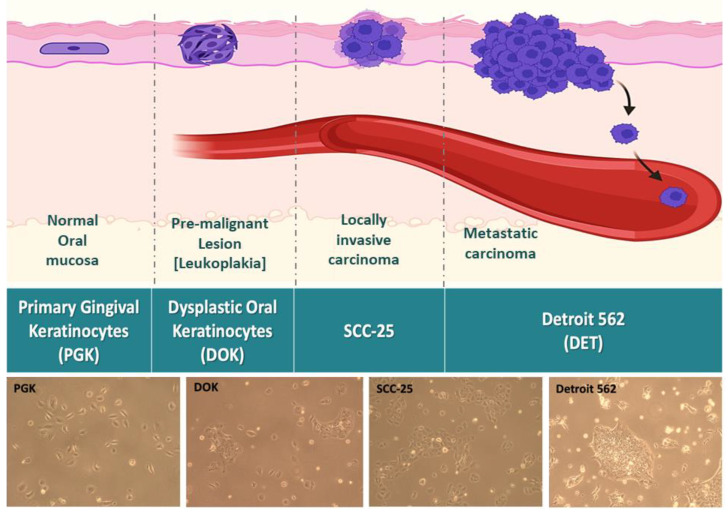
Schematic depicting four cell lines representing four stepwise progressive steps of OSCC, from normal health to dysplasia, then to locally invasive OSCC, and finally to metastatic carcinoma. The normal oral mucosa is represented by the cell line PGK. Leukoplakia is modeled using DOKs. Locally invasive carcinoma is modeled using SCC-25, whose growth rate is faster than that of PGKs or DOKs. Metastatic carcinoma is modeled using Detroit-562, whose growth and morphology differ greatly from those of the other three lines. Arrows indicate invasion and intravasation of metastatic cells.

**Figure 2 genes-16-00209-f002:**
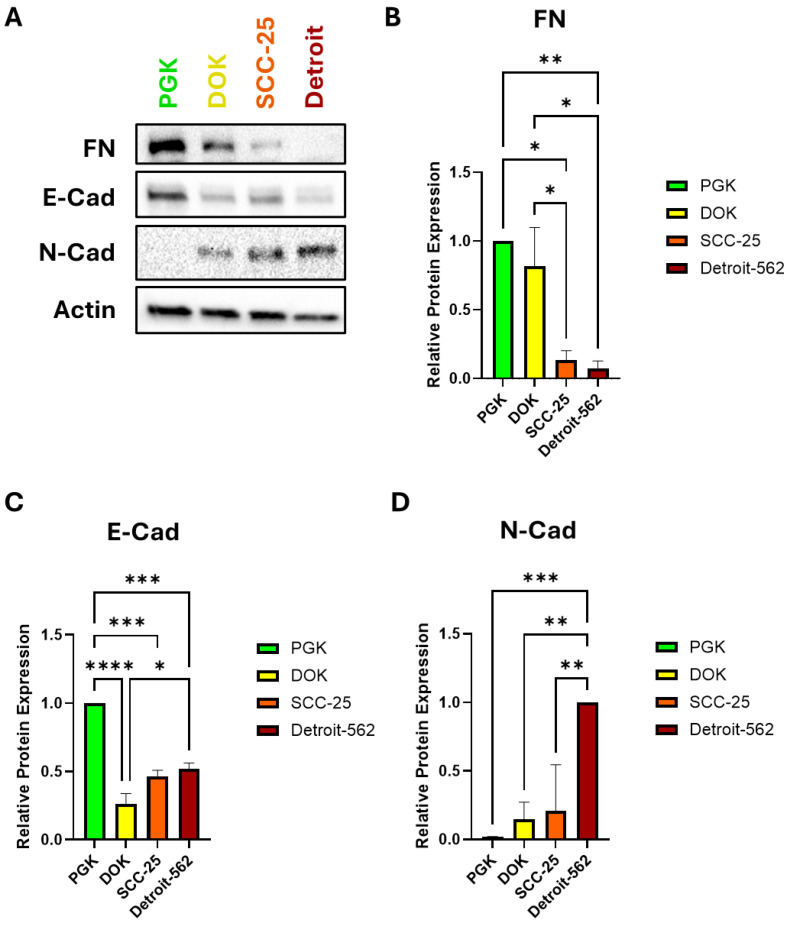
Protein expression of EMT markers. (**A**) Representative Western blot of fibronectin (FN), E-cadherin (E-Cad), and N-cadherin (N-Cad). Actin served as the loading control for all Western blot tests. (**B**) FN expression is significantly downregulated in SCC-25 and Detroit cells compared to PGKs and DOKs. (**C**) E-Cad is significantly downregulated in DOKs, SCC-25, and Detroit compared to PGKs. Expression rebounds slightly from DOKs to Detroit. (**D**) N-Cad expression trends higher with progression, with a statistically significant increase in the Detroit cells. N-Cad was difficult to detect at all in the normal PGK cells; hence, N-Cad was normalized to the expression in Detroit 562. * *p* < 0.05, ** *p* < 0.01, *** *p* < 0.001, **** *p* < 0.0001. *n* = 3 per protein.

**Figure 3 genes-16-00209-f003:**
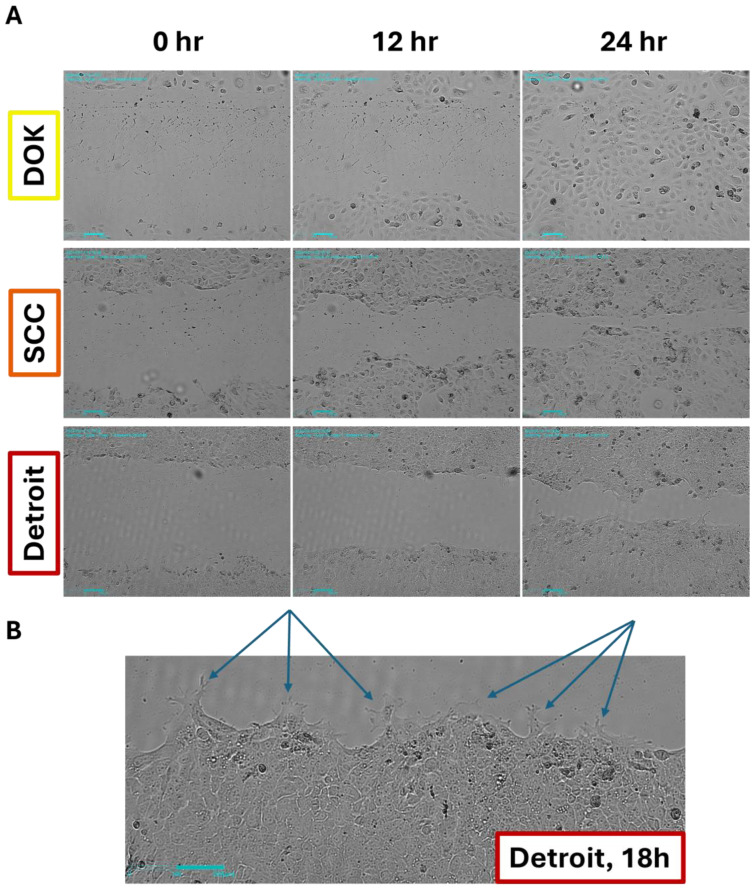
Wound healing assay reveals differences in the migratory phenotypes. (**A**). While all three lines show a capacity for wound healing, the Detroit cells show distinct projections and a leading edge of the migratory front that the other lines lack (**B**). This suggests Detroit relies more heavily on migration and DOKs more on proliferation to fill the gap. See also Appendix A. Scale bars: 200 µm.

**Figure 4 genes-16-00209-f004:**
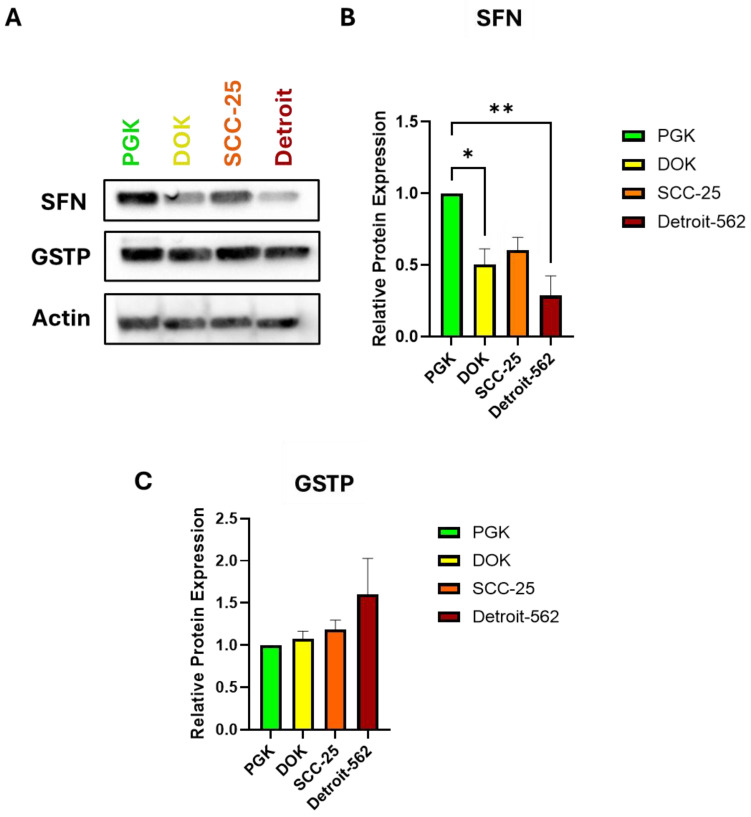
Protein expression of putative biomarkers of OCSS progression. (**A**) Representative Western blot of stratifin (SFN) and GSTP in all four cell lines. Actin serves as the loading control. (**B**) Quantification of SFN. DOKs and SCC-25 show similar SFN levels but both have lower levels than those in PGKs, and the expression in Detroit is lower still. Comparisons of each pair of steps were analyzed using an ANOVA. (**C**). Quantification of GSTP. An upward trend was seen with progression but did not achieve statistical significance. * *p* < 0.05, ** *p* < 0.01, *n* = 3 independent experiments.

**Figure 5 genes-16-00209-f005:**
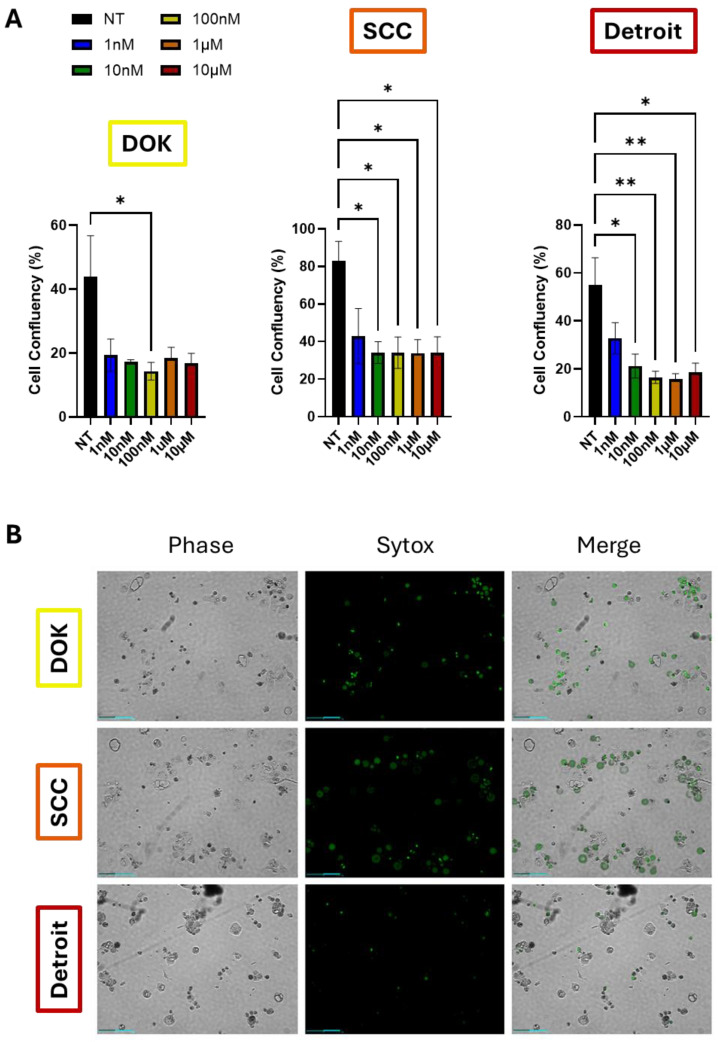
Analysis of the drug responses in OSCC cell lines. (**A**) Celloger measurements of the confluency of the DOK, SCC-25, and Detroit 562 cells 72 h after treatment with the indicated doses of docetaxel. All of the graphs depict the mean ± SEM from three independent experiments, with each being the average of 3 technical replicates. The growth curves over the full 72 h treatment window are shown in Appendix A. * *p* < 0.05, ** *p* < 0.01. (**B**) Representative images of DOK, SCC-25, and Detroit cells following 72 h of exposure to docetaxel (10 µM). Sytox green dye marks dead cells. Detroit cells exhibit less Sytox staining, suggesting growth inhibition rather than toxicity, as seen in DOKs and SCC-25. Scale bars: 200 µm.

**Figure 6 genes-16-00209-f006:**
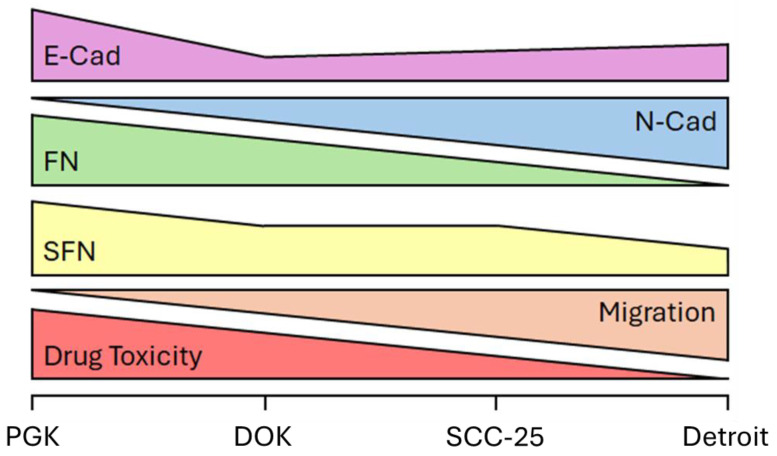
Schematic of our main findings. A cadherin switch is accompanied by a loss of FN and SFN expression as OSCC progresses. Meanwhile, the ability of the cells to migrate and their susceptibility to docetaxel were inversely correlated.

## Data Availability

All of the data are contained within the article or the Appendix A available accompanying the article online.

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
