# Peer review of "Putative Biomarkers for Prognosis, Epithelial-to-Mesenchymal Transition, and Drug Response in Cell Lines Representing Oral Squamous Cell Carcinoma Progression"

_genes, 2025, doi:10.3390/genes16020209_

Round 1
Reviewer 1 Report
Comments and Suggestions for Authors
In this study, the authors attempted to identify molecular biomarkers which enables to predict the prognosis of oral squamous cell carcinoma (OSCC). By using multiple cell lines which mimic malignant progression of OSCC, they demonstrated that epithelial-mesenchymal transition (EMT) is the key mechanism to drive OSCC progression and drug resistance. This study is well designed, and experimental results are solid to support the conclusion. I suggest several minor issues which need to be addressed before acceptance.
Page 4, line 140: In the method of western blotting, the authors should describe how they quantified the results.
Page 6, line 206: In Figure 2D, it is unclear why the authors could draw error bars even though there are only 2 replicates in the experiments.
In Figure 3, the results of wound healing results should be quantified.
In Figure 4, why the authors used GAPDH as an endogenous control for SFN, while they used Actin for GSTP?
Page 9, line 261: The description “DOK 261 and SCC-25 cells both showed the same reduction” is not accurate, because although the degrees of reduction in cell growth and proliferation were similar among different doses, they were not completely the same.
Reviewer 2 Report
Comments and Suggestions for Authors
In the manuscript presented, the Authors examine putative biomarkers for prognosis, epithelial to mesenchymal transition, and drug response in cell lines representing oral squamous cell carcinoma progression. The introduction and discussion are engaging and well-written, making the article enjoyable. However, I have a few comments:
1. Genes symbols should be in italics.
2. Could the Authors clarify why they chose the Detroit 562 cell line for their study? This cell line is associated with oropharyngeal squamous cell carcinoma (OPSCC), primarily affecting the base and posterior one-third of the tongue, the tonsils, the soft palate, and the posterior and lateral pharyngeal walls. Therefore, the location differs from oral squamous cell carcinoma (OSCC).
3. I recommend changing the term "oropharyngeal cancer" to "oral squamous cell carcinoma” (lines 122-123).
4. The supplementary files do not contain tables and videos.
5. I recommend including information in the "Western Blotting" section that GAPDH and Actin were utilized as endogenous control and justifying this choice.
6. The results section must not include data from other publications; this information should be presented in the discussion section.
